# Long Noncoding RNA VLDLR-AS1 Levels in Serum Correlate with Combat-Related Chronic Mild Traumatic Brain Injury and Depression Symptoms in US Veterans

**DOI:** 10.3390/ijms25031473

**Published:** 2024-01-25

**Authors:** Rekha S. Patel, Meredith Krause-Hauch, Kimbra Kenney, Shannon Miles, Risa Nakase-Richardson, Niketa A. Patel

**Affiliations:** 1Research Service, James A. Haley Veteran’s Hospital, 13000 Bruce B Downs Blvd., Tampa, FL 33612, USA; rekha.patel1@va.gov (R.S.P.); shannon.miles@va.gov (S.M.); 2Department of Molecular Medicine, University of South Florida, Tampa, FL 33612, USA; meredithk@usf.edu; 3Department of Neurology, Uniformed Services University of the Health Sciences, Bethesda, MD 20814, USA; kimbra.kenney@usuhs.edu; 4Department of Psychiatry & Behavioral Neurosciences, Morsani College of Medicine, University of South Florida, Tampa, FL 33620, USA; 5Chief of Staff Office, James A. Haley Veteran’s Hospital, Tampa, FL 33612, USA; risa.richardson@va.gov; 6Department of Internal Medicine, Pulmonary, Critical Care and Sleep Medicine, University of South Florida, Tampa, FL 33620, USA

**Keywords:** mild traumatic brain injury, lncRNA, lincRNA, exosomes, VLDLR-AS1, MALAT1, depression, LIMBIC CENC, neurological diseases, neuropsychological symptoms, genetic serum biomarker, veterans, PTSD

## Abstract

More than 75% of traumatic brain injuries (TBIs) are mild (mTBI) and military service members often experience repeated combat-related mTBI. The chronic comorbidities concomitant with repetitive mTBI (rmTBI) include depression, post-traumatic stress disorder or neurological dysfunction. This study sought to determine a long noncoding RNA (lncRNA) expression signature in serum samples that correlated with rmTBI years after the incidences. Serum samples were obtained from Long-Term Impact of Military-Relevant Brain-Injury Consortium Chronic Effects of Neurotrauma Consortium (LIMBIC CENC) repository, from participants unexposed to TBI or who had rmTBI. Four lncRNAs were identified as consistently present in all samples, as detected via droplet digital PCR and packaged in exosomes enriched for CNS origin. The results, using qPCR, demonstrated that the lncRNA VLDLR-AS1 levels were significantly lower among individuals with rmTBI compared to those with no lifetime TBI. ROC analysis determined an AUC of 0.74 (95% CI: 0.6124 to 0.8741; *p* = 0.0012). The optimal cutoff for VLDLR-AS1 was ≤153.8 ng. A secondary analysis of clinical data from LIMBIC CENC was conducted to evaluate the psychological symptom burden, and the results show that lncRNAs VLDLR-AS1 and MALAT1 are correlated with symptoms of depression. In conclusion, lncRNA VLDLR-AS1 may serve as a blood biomarker for identifying chronic rmTBI and depression in patients.

## 1. Introduction

Traumatic brain injury (TBI) accounts for 30% of all injury-related fatalities. An estimated 2 million people suffer from TBI annually in the United States, and worldwide, an estimated 57 million people have been hospitalized with one or more TBIs [1,2]. The Centers for Disease Control and Prevention (CDC) estimated that annually, 1.7 million Americans survive TBI. TBI can result in long-lasting alterations in brain function and genome-wide changes in gene expression [3]. Older adults and military personnel have higher incidences of TBI compared to other civilians [4]. TBI incidences in military personnel serving in war conflicts have increased by more than 35% since the global war on terrorism (GWOT) [5]. Mental health comorbidities that occur alongside TBI negatively impact our service members’ or veterans’ ability to return to a normal independent life and are associated with higher health service utilization and costs to the Veterans Affairs hospitals across the nation [6,7]. The Department of Defense (DoD) and the Traumatic Brain Injury Center of Excellence (TBICoE) statistics indicate that among the US service members with TBI, the majority (82.3%) have sustained a mild TBI (mTBI) [8]. Moderate and severe TBIs have pronounced clinical outcome measurements; however, individuals with mTBI often do not present with dramatic clinical abnormalities after the acute phase of injury. While the majority of service members recover within three months of mTBI, 10–15% of individuals who have deployment-related repetitive mTBI (rmTBI) experience chronic post-concussive symptoms, such as headaches or dizziness, beyond this timeframe [9].

The chronic effects of rmTBI may also include lower cognitive ability, depression or a decrease in the quality of life [10]. Fewer than 10% of patients with mTBI seek care for these symptoms in the following years [11,12]. While several tools, such as self-administered questionnaires and psychological tests, are available to measure neurobehavioral performances [13], it is difficult to evaluate whether changes in neurological behavior are related to the rmTBI events or occur independently due to other personal psychological traumas [14,15]. According to the updated Lancet Neurology Commission’s report of 2022 [16], there is a growing interest in identifying patients who have a history of mTBI and who will benefit from personalized treatment plans during their long-term care. In addition to CT scans [17,18], a recent study by Vedaei et al. described the use of functional MRI and machine-learning techniques to distinguish patients with chronic mTBI from healthy individuals with no TBI incidences [19]. However, these imaging techniques are costly and time-consuming, thereby dramatically decreasing their use in screening all patients. The development of quantitative blood-based biomarkers to identify chronic rmTBI patients is a dire and current need [16].

The analysis of blood for potential mTBI biomarkers is advantageous compared to analyzing CSF, as it is not as invasive as collecting CSF. The collection of blood samples from patients is a regular task at a clinic, and blood samples can be collected on an annual basis. Proteins such as ubiquitin C-terminal hydrolase-L1 (UCH-L1), S100 calcium-binding protein B (S100B), glial fibrillary acidic protein (GFAP), cytosolic tau (c-tau) and myelin basic protein (MBP) have been suggested as blood biomarkers for moderate and severe TBI [20,21,22]. These proteins are released with the rupture of the brain membranes after injury, and their levels are highest in the acute phases of injury. However, mTBI does not cause severe damage, and many proteins are unable to cross the blood–brain barrier (BBB) efficiently, which hinder their consistent detection in blood. A recent study showed that GFAP, c-Tau and S100B were not reliable biomarkers for mTBI [23]. Previous studies have shown that exosomes derived from the central nervous system (CNS) can cross the BBB and are detected in the blood. Exosomes carry a cargo of long noncoding RNA (lncRNA) and miRNA, and their levels within the circulating exosomes reflect the state of neurodegenerative diseases [24]. The human genome comprises ~2% coding genes, while ~98% of genes are noncoding, emphasizing the importance of these non-coding transcripts in cellular function. LncRNAs are master regulators of gene expression, functioning by regulating transcription, splicing, chromosome looping, chromatin modification, mRNA decay, as well as the regulation of miRNA in normal and disease states [25,26,27,28]. Levels of lncRNA detected in bodily fluids, such as blood, saliva and urine, are shown to be indicative of disease states and are suggested as biomarkers for cancer, bipolar disorder and COVID-19 [29,30,31,32,33,34]. The advantages of using lncRNA as biomarkers include their abundance and stability in body fluids, specificity to a disease, detectability using common molecular biology methods and reproducible quantitative results [35,36,37,38,39].

This study sought to determine a lncRNA expression signature in serum samples that identifies individuals with a history of rmTBI from those with no lifetime TBI events in US veterans. The serum samples were obtained from the Long-Term Impact of Military-Relevant Brain-Injury Consortium Chronic Effects of Neurotrauma Consortium (LIMBIC CENC), which is a multi-center collaboration involving clinicians and researchers from the DoD, VA, as well as academic and research institutes. LIMBIC CENC seeks to fill the gaps in knowledge about the basic science of mTBI, determine the late-life outcomes after mTBI and develop effective treatment strategies. A structured clinical interview, medical record review, and expert consensus are used to diagnose a history of mTBI. Along with the blood samples, the clinical data from LIMBIC CENC are collected as part of a prospective, longitudinal study of mTBI and are evaluated for neuropsychological symptom burden. Hence, a second goal was to evaluate the levels of lncRNA for correlation with psychological symptom burden, such as cognition, memory, depression or post-traumatic stress disorder (PTSD), concomitant with rmTBI in the participants. Our results demonstrate that circulating lncRNA very low-density lipoprotein receptor-antisense 1 (VLDLR-AS1) levels are negatively corelated with rmTBI and can be used to identify individuals with rmTBI independent of the number of years elapsed since the TBI events. Additionally, lncRNA VLDLR-AS1 and MALAT1 levels are correlated to depression states in the participants. In summary, VLDLR-AS1 may be used a blood biomarker for identifying patients with chronic rmTBI who may benefit from personalized long-term care.

## 2. Results

### 2.1. Identifying lncRNAs Consistently Detected in Serum

The participant demographics and characteristics are shown in Table 1. Serum samples were obtained from the LIMBIC CENC repository (as described in Section 4.1) from US veterans who were either unexposed to TBI (24 participants) or had rmTBI (rmTBI: >2 and up to 10 rmTBI incidences; 43 participants) with at least 9 years elapsed since the last TBI and up to 20 years since their first TBI. The rmTBI group consisted of combat-associated mTBI and non-combat mTBI events. The mean age of the participants was 41 years, and there were equal percentages of gender, age and racial background between the no TBI and rmTBI cohorts. There were more male participants (79.1%) compared to females, which follows the overall trend of the percentage of women in active-duty force (Department of Defense, annual report on demographics 2022). Within the participants, the majority (66–76%) self-reported as white, while 23–33% reported their racial and ethnic background as black or African American. There were no Asians, American Indians or Native Hawaiians in this cohort of participants.

First, to identify which lncRNAs were consistently detected across all serum samples, we evaluated randomly chosen samples (no TBI: n = 5 and rmTBI: n = 10) using RT-qPCR. Total RNA was extracted, and cDNA was synthesized. For screening the lncRNAs, the cDNA from the no TBI group was pooled, and the cDNA from the rmTBI group was pooled separately. These pooled samples were evaluated using the human lncRNA profiler qPCR array (SBI System Biosciences, Palo Alto CA, USA). The array includes functionally significant human lncRNAs, reference controls and internal assay controls, such as 18s rRNA, RNU43, LAMIN A/C, GAPDH and U6 (Figure 1a shows the human lncRNA profiler qPCR array plate layout). The Ct values are generated using qPCR amplification and normalized to the internal controls on the plate. The results are calculated as ΔΔCT and then plotted as fold changes on a logarithmic scale, representing a change in levels between the pooled rmTBI sample and the no TBI group. The fold change beyond 0.01 on the log graph indicates extremely low amounts of the target in samples and is, therefore, not conducive for clinical interpretation, and thus, is excluded. A fold change of 1 indicates no discernable difference between the groups in comparison and is therefore not considered a candidate to differentiate between the groups. The results indicated that the consistently detected lncRNAs in measurable quantities in serum samples were metastasis-associated lung adenocarcinoma transcript 1 (MALAT1), growth arrest specific 5 (GAS5), nuclear paraspeckle assembly transcript 1 (NEAT1) and very low-density lipoprotein receptor antisense RNA 1 (VLDLR-AS1) (Figure 1b).

We sought to verify the results of the qPCR array, which indicated that MALAT1, GAS5, NEAT1 and VLDLR-AS1 were detected in serum samples. Hence, we evaluated the number of gene copies of the lncRNAs MALAT1, GAS5, NEAT1 and VLDLR-AS1 in the individual serum samples (no TBI: n = 5 and rmTBI: n = 10) using the droplet digital PCR (ddPCR; described in Method Section 4). The ddPCR method provides an absolute count of the gene copy per mL of serum (Figure 1c). The ddPCR results determined that these four lncRNAs, MALAT1, GAS5, NEAT1 and VLDLR-AS1, were consistently present in the serum of the participants with no TBI exposure, as well as in serum from individuals with rmTBI.

### 2.2. LncRNAs Are Packaged in Exosomes Derived from the Brain

The lncRNAs detected in the serum are packaged within extracellular vesicles (EVs) to prevent their degradation. Exosomes are nano-sized EVs (30–150 nm in size) circulating in the serum, and specific cell surface markers indicate their cellular origin. We sought to evaluate whether the four identified lncRNAs in serum (shown in Figure 1a,b) were also present in the exosomes derived from the CNS. We used the L1CAM antibody (brain-specific protein) to purify brain-derived exosomes in pooled serum samples (10 each from no TBI and rmTBI serum, totaling 20 samples), followed by characterization of the exosomes using NanoSight (as described in the method section). RNA was then extracted from the exosomes, followed by the measurement of lncRNA levels using qPCR. The same pooled serum sample was evaluated for total amounts of lncRNAs from all sources using qPCR. The results demonstrate that, in addition to their presence in serum (measuring all sources of origin), the lncRNAs MALAT1, GAS5, NEAT1 and VLDLR-AS1 were detected in the brain-derived exosomes (Figure 2: absolute quantification (AQ) in ng). The results also indicate that VLDLR-AS1 was the most abundant lncRNA in circulation in serum compared to MALAT1, NEAT1 or GAS5. Additionally, the statistical analysis of the geometric mean with a 95% confidence interval (geo mean 95% CI: a measure of central tendency and quantities between groups (total versus brain-derived) occurring in a relative manner) shows that 20.5% of circulating VLDLR-AS1 was derived from the brain (VLDLR-AS1 95% total CI: 116.5, 285.6; brain: 23.89, 58.54). Further, the results showed that 13.9% of circulating GAS5 was derived from the brain (GAS5: 95% total CI: 52.26, 92.67; brain: 7.265, 12.88), 30.9% of circulating NEAT1 was derived from the brain (NEAT1: 95% total CI: 0.5716, 1.744; brain: 0.1766, 0.5389) and 24% of circulating MALAT1 was derived from the brain (MALAT1: 95% total CI: 7.569, 23.77; brain: 1.817, 5.704). The results demonstrate that the levels of these four lncRNAs in serum versus CNS-derived exosomes were consistent in all samples, irrespective of rmTBI or no TBI status. Hence, direct measurements of the lncRNA levels in serum can be used to evaluate any differences in the levels of lncRNAs between no TBI and rmTBI samples and would be indicative of the trend of CNS-secreted lncRNAs.

### 2.3. LncRNA VLDLR-AS1 Levels Are Lower in Participants with Repetitive mTBI

Since isolation and characterization of exosomes is a long process, and our results (Figure 2) showed a consistent percentage of these lncRNAs in serum versus exosomes, we focused on the direct measurement of lncRNAs in serum to enable an easier translation of the method to clinical settings. We measured the levels of these four identified lncRNAs in the serum samples of all 67 individual participants using qPCR. For this analysis, we compared the levels of lncRNAs MALAT1, GAS5, NEAT1 and VLDLR-AS1 between samples with no lifetime TBI and those with rmTBI. Total RNA was extracted from the serum samples, followed by the synthesis of cDNA. An amount of 1 μL of cDNA was used in SYBR Green qPCR for amplification, using primers specific to MALAT1, GAS5, NEAT1, VLDLR-AS1 and the housekeeping gene U6. A standard curve was generated for individual lncRNAs and U6 and used to calculate absolute quantities (AQs) of each lncRNA per mL of serum. The results determined that the levels of MALAT1, NEAT1 and GAS5 did not change significantly between the no TBI and rmTBI groups (Figure 3a). The levels of VLDLR-AS1 were lower in the rmTBI samples compared to the no lifetime TBI samples, and this difference was highly significant (*p* = 0.0004), indicating that VLDLR-AS1 is a strong, viable indicator of rmTBI. Hence, we further analyzed the correlation between VLDLR-AS1 levels in rmTBI.

Next, we sought to evaluate if the levels of VLDLR-AS1 changed with the number of years elapsed since the mTBI event. Our results determined that the levels of VLDLR-AS1 in serum did not statistically corelate with the number of years since the TBI event (*p* = 0.8) (Figure 3b). Taken together, the results show that low VLDLR-AS1 levels in serum are correlated with rmTBI incidents, independent of the number of years elapsed since the mTBI event.

A receiver operating characteristic (ROC) curve was generated (Figure 3c) for VLDLR-AS1 to evaluate the relationship between sensitivity and specificity regarding the status of rmTBI events. ROC analysis revealed an area under the curve (AUC) of 0.74 (95% CI: 0.6124 to 0.8741; *p* = 0.0012). The optimal cutoff for VLDLR-AS1 was less than or equal to 153.8 ng/mL of serum, measured as absolute quantities using qPCR, and this value had a sensitivity of 69.8%. These results demonstrate that a decrease in circulating lncRNA VLDLR-AS1 levels strongly correlates with the prevalence of rmTBI events and can be detected in samples measured years after the mTBI incidents.

In a two-sided Z-test for two means, the no TBI group has a sample size of 24, while the rmTBI group has a sample size of 43. This test achieves a power of 90.89% at the 5% significance level. The difference in sample means under the alternative hypothesis is 87.3, and the standard deviations of the no TBI group and the rmTBI group are 119.6 and 67.65, respectively, indicating that serum VLDLR-AS1 levels have a strong biomarker potential to identify rmTBI status in patients.

### 2.4. Secondary Analysis of the Clinical Data Shows VLDLR-AS1 and MALAT1 Levels Are Correlated with Depression

A secondary analysis of the clinical data of LIMBIC CENC was performed to evaluate the correlation between the psychological symptom burden and the identified lncRNAs detected in serum. We first analyzed the levels of the four lncRNAs (MALAT1, GAS5, NEAT1 and VLDLR-AS1) measured in serum (AQ: calculated using RT-qPCR) to correlate them with the Neurobehavioral Symptom Inventory-22 total score (NSI22_tot). The NSI22_tot is a self-administered 22-item questionnaire assessing post-concussion syndrome (PCS) symptoms that may emerge after mTBI. The four categories span somatosensory, cognitive, affective and vestibular responses to questions about PCS symptoms [40,41]. The normative data for NSI22_tot were calculated as described by [42,43] and were generated by LIMBIC CENC clinicians. Here, we evaluated whether the lncRNAs VLADLR-AS1, MALAT1, GAS5 or NEAT1 levels (AQ values) had a linear relationship with the normative NSI22_tot values, using the descriptive statistics Pearson correlation coefficient (bivariate correlation). A *p*-value less than 0.05 was considered statistically significant. The correlation coefficient (r value) of 0 denotes no relationship strength, a range from 0 to +0.3/−0.3 denotes a weak relationship, a range from +0.3/−0.3 to +0.5/−0.5 denotes a moderate relationship and a range from +0.5/−0.5 to >0.5/−0.5 denotes a strong relationship. The analysis shows that none of the lncRNAs had a *p* value of less than 0.05, and the r values indicated a weak relationship. Hence, we conclude that there is no significant correlation between the NSI22_tot (Figure 4a) and the MALAT1, GAS5, NEAT1 and VLDLR-AS1 levels.

Next, we evaluated the correlation between the PTSD checklist for DSM-5 (PCL-5) and the levels of these four lncRNAs. The PCL-5 is a self-report measure corresponding to symptoms of PTSD, and a cutoff score of 33 suggests PTSD in the individual. Data were sorted into a PCL-5 score less than 33 (no PTSD group) and a PCL-5 score equal to or greater than 33 (PTSD group) and plotted against the AQ levels of the lncRNA MALAT1, GAS5, NEAT1 and VLDLR-AS1. The *t*-test was used to analyze the significance of comparing the no PTSD and PTSD groups, with *p* < 0.05 considered significant. The results indicated (Figure 4b) no significance between the no PTSD group compared to the PTSD group for the four lncRNAs (MALAT1, GAS5, NEAT1 and VLDLR-AS1) levels.

Finally, the levels of lncRNAs VLDLR-AS1, NEAT1, GAS5 and MALAT1 were analyzed for correlation with depression symptoms, measured using the Patient Health Questionnaire-9 (PHQ-9) [44]. The PHQ9 self-reports were scored for depression severity by the LIMBIC CENC clinicians. Here, we analyzed whether the levels of the lncRNAs correlated with the depression severity groups: none, minimal to mild, moderately severe to severe. These groups were plotted against the AQ levels of MALAT1, GAS5, NEAT1 and VLDLR-AS1 and analyzed using an analysis of variance (ANOVA) statistical test. A *p*-value less than 0.05 is considered statistically significant. The analysis revealed that the levels of lncRNAs GAS5 and NEAT1 did not show a significant correlation with depression levels. The lncRNAs VLDLR-AS1 and MALAT1 changed significantly (*p* < 0.05) with respect to the depression levels. We further analyzed VLDLR-AS1 and MALAT1 levels, differentiating between the no TBI and rmTBI cohorts, as shown in Figure 4c. The results show that VLDLR-AS1 levels were lower in the rmTBI group compared to the no TBI cohort. There was a significant difference when comparing the groups with none to minimal–mild depression levels and when comparing the groups with minimal–mild to moderately severe–severe depression levels. The MALAT1 levels did not show significance when comparing between the no TBI and rmTBI groups, but they were significantly different when comparing the groups with none to minimal–mild depression levels and when comparing the groups with minimal–mild to moderately severe–severe depression levels.

## 3. Discussion

Individuals with rmTBI may have accompanying chronic sequalae, such as changes in cognition, PTSD and depression. We sought to identify if any quantifiable lncRNA biomarkers in body fluids could be indicative of mTBI, years after the mTBI incidents. Peripheral blood analysis is usually performed as part of routine clinical care, and hence, we analyzed the serum for discovering a lncRNA whose levels may indicate mTBI. LncRNAs are detected in blood (serum or plasma) and many studies have previously shown a correlation between the circulating levels of lncRNA and diseases such as diabetes and cancers [45,46,47,48,49,50,51]. LncRNAs are regulatory RNAs longer than 200 nucleotides without open reading frames, and hence, they do not code for a protein. Long intergenic noncoding RNAs (lincRNA) are a subset of lncRNAs, and their transcription units do not overlap with a protein-coding transcript [52]. Most publications do not differentiate between lincRNAs and lncRNAs, and these are grouped together as lncRNAs. LncRNAs are integral to the regulation of brain functions, as they control the expression of genes, thereby affecting the brain’s proteome, as well as neuronal transcription factors, chromatin modeling, splicing and post-translational modifications of protein [53,54,55]. Aberrant expression patterns of lncRNAs are indicative of disease states, including neurological diseases [56,57,58].

Here, we undertook a qPCR-based array approach to first establish which lncRNAs are consistently present in measurable quantities in the serum from US veterans. There is a limited number of lncRNAs on the array compared to the 5000+ lncRNAs detected across species; however, this approach was chosen, as the list comprises functionally established lncRNAs in humans. In the array results (Figure 1), lncRNAs with fold changes beyond 0.01 on the log scale were not included. This was necessary to avoid errors in measuring and detecting very low levels of lncRNAs, which hinder the reproducibility of results. We validated the results of the array with a secondary method using ddPCR, which is a new method to quantify genetic material. The ddPCR method incorporates the division of each sample into 20,000 nanodroplets, followed by the amplification of each droplet through PCR. Since this technology requires state-of-the-art equipment and highly trained personnel, the ddPCR method would not be easily incorporated into a clinical setting. Additionally, the ddPCR results for copy numbers were in concurrence with the the qPCR results, showing that the VLDLR-AS1 gene copy number is lower in rmTBI compared to the no TBI group. Hence, we used the qPCR method to evaluate the complete set of serum samples and quantify the levels of the lncRNAs VLDLR-AS1, MALAT1, NEAT1 and GAS5 with the goal of identifying a blood biomarker to distinguish samples with no lifetime TBI from those with rmTBI.

Our results demonstrated that the levels of VLDLR-AS1 (also referred to as linc-VLDLR and lncRNA VLDLR-AS1) in serum were significantly different between those with rmTBI and those with no TBI. ROC analysis revealed an area under the curve (AUC) of 0.74 (95% confidence interval (CI): 0.6124 to 0.8741; *p* = 0.0012). Power analysis comparing sample sizes for the no TBI group and rmTBI achieved 90.89% power at the 5% significance level. In addition, these samples were from patients who were clinically diagnosed with rmTBI. Taken together, our results demonstrate that serum VLDLR-AS1 levels are a strong indicator of the rmTBI status in patients.

VLDLR-AS1 was previously shown to contribute to cellular stress response in hepatocellular cancer [59]. High levels of VLDLR-AS1 in blood are associated with ovarian cancer [60] and fat reduction in cancer cachexia [61]. VLDLR-AS1 is expressed in the brain (Harmonizome 3.0 [62,63]), but its function in brain pathology or its role in the mechanism of brain injury is not known. VLDLR-AS1 is transcribed as an antisense RNA on the opposite strand of the VLDLR gene. Though such naturally occurring antisense RNA may modulate the expression of the sense gene, as in the case of HMGA2 [64] in pancreatic cancer, currently, it is unknown if VLDLR-AS1 affects the function of VLDLR in the brain. Gene-trait associations in the brain show that VLDLR-AS1 is associated with neurological diseases (as indicated by the dbGAP gene-trait association dataset). Future research is needed to establish the function of VLDLR-AS1 in the brain and TBI. This could involve bioinformatics-based network analysis and biochemical array screening of brain tissue from rmTBI versus no TBI samples to identify target genes and pathways of VLDLR-AS1. A similar screening approach, followed by in vitro determination of the underlying function, was previously undertaken by us to understand the role of the lncRNA GAS5 and identify the insulin receptor as its target [50,65].

A retrospective chart review study by Powell et al. [66] investigated the accuracy of mTBI diagnosis via the ICD-9-CM (International Statistical Classification of Diseases, 9th Revision, Clinical Modification) codes, as defined by the CDC. Over 50% of the patients who met the CDC criteria for mTBI were not diagnosed during their hospital stay, resulting in a delay or prevention of appropriate care. Mild TBI is commonly considered difficult to diagnose if the patient’s history is not available, mostly because there is no universally agreed-upon clinically diagnostic criteria for mTBI that discriminate between mTBI and other mental health conditions [67,68,69]. Thus, by measuring serum VLDLR-ASI levels in suspect patients at admission and over several years during routine check-ups, the proper treatment can be provided in a time-sensitive manner, improving patient outcomes.

Other proteins, such as GFAP, UCH-L1, MBP are proposed as biomarkers of moderate and severe TBI; however, their specificity in the determination of rmTBI years after the incidents is not evaluated. A previous study using the LIMBIC CENC samples showed an increase in total and phosphorylated tau in exosomes, which was measured using an immunoassay conducted on rmTBI plasma samples [70]. This approach requires immunoassays, which take up to 2 days to complete, while the measurement of lncRNAs can be performed via qPCR in 2 h, significantly enabling translation into routine patient care. Previous research from our laboratory and others has shown that the lncRNA VLDLR-AS1 is packaged in exosomes [59,71,72]. The sources of VLDLR-AS1 in the exosomes in serum may originate from other organs too. Here, our results demonstrated that the VLDLR-AS1 levels measured in total serum are consistently proportionate to the levels found in CNS-derived exosomes circulating in the blood in both cohorts of rmTBI and no TBI. This indicates that measurements of VLDLR-AS1 in the serum reflect the changes in levels of brain-derived VLDLR-AS1. However, it is not yet known if there is any direct regulation between VLDLR-AS1 and tau or the packaging of this cargo together in the CNS-derived exosomes. This can be pursued in future studies using depletion experiments, evaluating the packaging of exosomal cargo in the brain and its secretion after mTBI.

While serum lncRNA biomarkers for rmTBI have not been described previously, the role of lncRNAs in brain injury has been demonstrated. MALAT1, NEAT1, SNHG1, MEG3, HOTAIR were shown to be involved in inflammation angiogenesis, apoptosis and cell proliferation pathways in the brain, and their levels changed in the brain tissue in mouse models of TBI [73,74,75,76,77,78]. These studies did not measure circulating blood levels of lncRNA in mouse models of TBI. Moreover, VLDLR-AS1 has not been evaluated in TBI pathology in these models. Another aspect to note is that though the functional role of lncRNAs is usually maintained across mouse and human lncRNAs due to the conservation of secondary structures, the sequences differ by up to 60% between mice and humans [79,80]. In scenarios where lncRNAs are detected in animal models of disease, the results are often not translated to human samples, as the primers used in animal studies may not recognize the human transcript for amplification via qPCR. Hence, our measurements in patient samples are highly relevant, as the findings can be directly translated to clinical settings.

Individuals who have endured mTBI, and particularly those with rmTBI, often have concomitant long-term neurological symptoms, including a decline in cognition, motor abilities and changes in behavior [81,82,83,84,85]. Extensive neuropsychological testing procedures have been established to evaluate the long-term effects, which include clinical assessments, as well as self-administered questionnaires, such as NSI-22 and PHQ-9 [86,87,88]. We conducted a secondary analysis of the psychological symptom burden in the LIMBIC CENC cohort to evaluate whether the levels of the four serum lncRNAs (VLDLR-AS1, MALAT1, NEAT1, GAS5) correlate to symptom burden (such as cognition, memory, depression or PTSD) after rmTBI. Interestingly, the levels of VLDLR-AS1, along with MALAT1, were correlated with depression in the participants. Using a genome-wide association study (GWAS), the VLDLR-AS1 levels were previously shown to be associated with the gambling trait [89]. However, the role of VLDLR-AS1 in depression is not yet clear, and further research is indicated. MALAT1 is shown to regulate survival pathways in animal models of brain injury [90,91]. Circulating MALAT1 has been studied as a biomarker for non-small cell lung cancer, epithelial ovarian cancer, hepatocellular carcinoma and tongue squamous carcinoma [92,93,94,95], though its association with depression is not yet known. Electroencephalography (EEG) measurement of gamma and theta bands [96] and presence of cytokines [97] are proposed as biomarkers to help with the diagnosis of depression. Our finding suggests that the serum lncRNA levels of VLDLR-AS1 and MALAT1 may provide a measurable, quantitative tool to screen for depression.

Blood-based lncRNA biomarker approaches provide a non-invasive method for the diagnosis, prognosis, as well as the evaluation of responses to clinical treatments. To translate the use of our finding into clinical practice, a premade qPCR plate specific for VLDLR-AS1 amplification may be provided to clinical laboratories. The serum samples can be added to the plates (direct qPCR) and run on RT-qPCR machines, requiring minimal technical expertise. The COVID-19 pandemic promoted the incorporation of RT-qPCR equipment in almost all clinical laboratories. The results of PCR could then be incorporated with other clinical findings, such as the patient’s history of brain injury and neuropsychological symptoms, to provide a robust diagnosis of mTBI. To establish VLDLR-AS1 as a stand-alone blood biomarker to identify patients with rmTBI, a validation study should be performed in the future in a larger population, including veterans and civilians with rmTBI and encompassing a broad range of age, gender and ethnicity. In a recent review, Arriaga-Canon et al. have described the sample size calculation for large validation studies, along with the clinical utility of lncRNAs as biomarkers in diseases [98]. A future validation study for VLDLR-AS1 will be started, with conditional power calculations for sample size determination and the establishment of the clinical utility of VLDLR-AS1 in the diagnosis, as well as prognosis, of rmTBI. Incorporating VLDLR-AS1 as a blood biomarker in annual check-ups will be beneficial to efficiently screen patients for lifetime mTBI incidences.

The results from our study support the notion that mTBI sequelae are chronic and genetic signatures that can be detected years after the injury. We demonstrate that lower levels of lncRNA VLDLR-AS1 in blood correlates with chronic rmTBI, which has been unknown thus far. These results will form the basis of future mechanistic studies to understand the role of VLDLR-AS1 in the brain, its packaging in exosomes and its secretion in the blood after mTBI. To evaluate VLDLR-AS1 as a robust prognostic biomarker for rmTBI and its treatment, longitudinal studies should also be undertaken, including the analysis of blood immediately after the first mTBI, after additional mTBI incidents and annual blood analysis for 10 years.

Broadly, the outcome of this study is also relevant to the civilian population, where mTBI occurs in sports, falls or accidents, and the individual often experiences neurological symptoms years after the injury. The integration of lncRNA-based tests into the diagnosis and prognosis of diseases is rapidly gaining importance due to the streamlined, simple method of qPCR, which offers high specificity, cost-effectiveness compared to other clinical tests and quick turnaround times (about 2 h). Clinical trials involving lncRNA biomarkers should be undertaken to move the field ahead and integrate their use in routine clinical diagnosis and patient care.

### Limitations

This was a discovery study to identify a lncRNA whose serum levels correlated with rmTBI. The limitation of the study is the evaluation of serum from only veteran population. This can be addressed by undertaking a similar study with a larger number of participants from a wider population, including civilians with rmTBI. In this study, the cohorts contain more males than females (due to the makeup of the armed forces), and in the future more samples should be evaluated with equal males and females, across different ages and race groups. None of the participants had any form of cancer, and the BMI was similarly distributed between the no TBI and rmTBI groups. However, it is possible that, while undertaking a future study with a larger number of participants, metabolic diseases and comorbidities may confound the results, as previous studies have shown changes in circulating lncRNA levels in diseases. These can be controlled by using a multi-logistic regression model, which can be evaluated for false discovery rates, and Bonferroni correction can be applied if needed. The model can be selected using goodness-of-fit and deviance statistics. Additionally, the serum obtained from participants immediately following an mTBI event may differ due to the body’s acute response to injury, involving inflammation and repair pathways. In the future, with more participant samples, variables found to be imbalanced across the compared groups (confounders) can be matched across the groups (mTBI and TBI negative, and separately, repetitive mTBI and single mTBI).

## 4. Materials and Methods

### 4.1. Study Samples

Participants were enrolled in the Long-Term Impact of Military-Relevant Brain-Injury Consortium Chronic Effects of Neurotrauma Consortium (LIMBIC CENC), a longitudinal multi-center study of mild TBI (mTBI) among post 9/11 era veterans and service members with combat exposure. As described previously [70], participants in LIMBIC CENC Study 1 were recruited from four Veterans Affairs medical centers who were combat-deployed, suffered possible concussive events, and were diagnosed with the spectrum of mTBI exposures (no lifetime events to repetitive combat-related events). The LIMBIC CENC study investigators conducted a structured interview, and TBI diagnosis was documented. The exclusion criteria, as described before included: (1) history of moderate or severe TBI, as defined by either (a) initial Glasgow Coma Scale < 13, (b) coma duration > 0.5 h, (c) PTA duration > 24 h or (d) traumatic intracranial lesion on head computerized tomography; or (2) history of (a) major neurologic disorder (e.g., stroke, spinal cord injury), (b) major psychiatric disorder (e.g., schizophrenia) with major defined as resulting in a significant decrement in functional status or loss of independent living capacity. None of the participants had any form of cancer noted. Serum was collected along with the participant’s history of mTBI exposures. A sample size of 60 samples in total was used (no TBI n = 20, rmTBI 2–3, n = 20 and rmTBI > 3, n = 20). The study achieved an 86% detection rate for a difference of 0.04 in the area under the ROC curve. The null hypothesis assumed an area under the ROC curve of 0.81, while the alternative hypothesis assumed an are of 0.84. The detection was performed using a two-sided Z-test at a significance level of 5%. We obtained 75 samples (25/group—a larger number of samples than the calculated 20/group) to start the analysis. However, equipment failure resulted in loss of 8 serum samples, and hence, for this study, we have a total of 67 samples, which is larger than the number calculated for achieving statistical significance. In the analysis, we combined all the rmTBI groups (number of rmTBI episodes 2–10) into a single group (n = 43) and compared it to the no TBI group (n = 23). The 67 serum samples from the biorepository and accompanying clinical data were provided through an MTA between James A. Haley Veterans Hospital and USUS, under an approved IRB (#Pro000173855).

### 4.2. Real-Time Quantitative Polymerase Chain Reaction (RT-qPCR)

Total RNA was isolated from serum using RNAzol RT reagent. A quantity of 1μg of RNA (260/230 > 1.8 and 260/290 > 1.8) was used to synthesize cDNA using ReadyScript^TM^ synthesis mix (Sigma RDRT, Burlington, MA, USA). The cDNA was then used either with the lncRNA profiler array (SBI System Biosciences, Palo Alto, CA, USA; catalog # RA900A-1) or in qPCR. Real-time qPCR was performed in triplicate using 1 μL of cDNA and Maxima SYBR Green/Rox qPCR master mix (Thermo Scientific, Waltham, MA, USA). Amplification was performed on the ViiA 7 (ABI). Primers were purchased from Origene (qSTAR qPCR primer pairs). These primer pairs were pre-designed, validated and tested for specificity to eliminate off-target effects. Primers had a Tm of 60–61 and an amplicon of 95–140 bp. The optimal primer concentration was determined from a range of 50–900 mM. The final concentration of each primer pair was selected to ensure efficiency and specificity for its target in serum based on the dissociation curve that showed a single, sharp peak, indicating that the primers amplify one specific target (described by us previously in [99]). Primers used in qPCR included GAS5 S 5′-CTTCTGGGCTCAAGTGATCCT-3′, GAS5 AS 5′-TTGTGCCATGAGACTCCATCAG-3′, MALAT1 S 5′-CTTCCCTAGGGGATTTCAGG-3′, MALAT1 AS 5′-GCCCACAGGAACAAGTCCTA-3′, NEAT1 S 5′-TCGGGTATGCTGTTGTGAAA-3′, NEAT1 AS 5′-TGACGTAACAGAATTAGTTCTTACCA-3′, VLDLR-AS1 S 5′-CAAGGATGGCAGTGATGAGGTC-3′, VLDLR-AS1 AS 5′-CTCGGATACCATTACACTGCCTG-3′, U6 S 5′-CGCTTCGGCAGCACATATAC-3′ and U6 AS 5′-TTCACGAATTTGCGTGTCAT-3′; GAPDH S 5′-GATCATCAGCAATGCCTCCT-3′ and GAPDH AS 5′-TGTGGTCATGAGTCCTTCCA-3′. For absolute quantification using SYBR Green qPCR, a standard curve was generated for each gene. For this, 100–0.4 ng of RNA from serum was reverse-transcribed, as described above. The resulting cDNA was used to obtain a standard curve correlating the amounts with the threshold cycle number (Ct values). A linear relationship (r^2^ > 0.96) was obtained for each gene. Plate set-up included the standard series, no template control and no reserve transcriptase control. The dissociation curve was analyzed for each sample. Absolute quantities (AQs) for expression levels of individual transcripts were calculated by normalizing the values to U6. AQs per mL of serum were then calculated. In addition, to validate the qPCR data, 1 µL of cDNA was amplified with the primer pairs, including GAPDH and U6 endogenous control primers, using JumpStart ReadyMix (Sigma P0982). PCR products were run on a 1% agarose gel and stained with ethidium bromide for visualization of bands and imaged in ProteinSimple FluorChem M. Primers were verified by visualizing a single band per sample, followed by densitometric analysis of the bands using integrated AlphaView^®^ software 3.5.0 (ProteinSimple, San Jose, CA, USA).

### 4.3. Droplet Digital Polymerase Chain Reaction (ddPCR)

Gene copy counts of VLDLR-AS1, GAS5, MALAT1 and NEAT1 were determined by housekeeping-independent measurement using ddPCR (Bio-Rad Laboratories, Hercules, CA, USA), which combines water-oil emulsion technology with microfluids. Each primer was optimized using a diluted 1:100 cDNA solution made from 500 ng of RNA. Droplets were generated using QX200 Droplet Generator (Bio-Rad Laboratories, Hercules, CA, USA), which transitions each sample into 20,000 nanoliter-sized droplets. The droplets are mixed with QX200 ddPCR EvaGreen Supermix to partition DNA and amplify it with the specific primers. The plate is then placed in the QX200 droplet reader to analyze each droplet individually. Results are analyzed using QuantaSoft™ Analysis Pro version 1.7.4.0917 (Bio-Rad) in 2D amplitude mode. Thresholds for positive and negative droplets are set up for BS analysis, and automatic thresholding determines concentrations and gene copy numbers. The reference copies (copies per 20 μL reaction) were exported into Microsoft Excel for further analysis, as previously calculated by [100]. The amount of input RNA (total RNA obtained from 1 mL serum) was obtained, and the theoretical RNA used per sample was calculated. The final results were calculated as copies per mL of serum.

### 4.4. CNS-Derived Exosome Purification

Exosomes were isolated from serum using the ExoCap kit from MBL. The serum was centrifuged at 10,000× *g* at 4 °C for 30 min to remove cellular debris. An equal volume of treatment buffer was added to minimize non-specific binding. To enrich CNS-derived exosomes, the L1CAM antibody was used with the ExoCap Streptvidin magnetic beads kit (MBL catalog # MEX-SA) and incubated at 4 °C for 18 h. The supernatant was removed, and the beads were washed with the washing buffer (from the kit). ExoCap nucleic acid elution buffer was used to collect the exosomes, followed by isolation of RNA. NanoSight (NTA3.1, Build 3.1.46 RRID SCR-014239) was used to analyze the purified exosomes. Analysis showed the exosomes size to be 97 ± 17 nm.

### 4.5. Neuropshycological Symptom Measures

The clinical data were provided by LIMBIC CENC and accompanied the serum samples. Post-concussive symptom severity was assessed using the Neurobehavioral Symptom Inventory (NSI) [101]. The NSI is a 22-item assessment with a three-factor structure (somatic/sensory, affective and cognitive), with a higher total score indicating a greater symptom burden. The Patient Health Questionnaire Depression Scale (PHQ-9) [44] was used to measure symptoms of depression, with higher scores indicating greater symptom severity. PCL-5 is a 20-item questionnaire corresponding to the DSM-5 symptom criteria for PTSD [102]. It is a self-report measure, and a total symptom severity score can be obtained by adding scores of individual items (with a range of 0–80). A provisional PTSD diagnosis can be made and a PCL-5 cutoff score of 31–33 is indicative of probable PTSD.

### 4.6. Statistical Analyses Methods

Descriptive statistics for all demographic and clinical variables were calculated using GraphPad Prism SPSS Analysis Software V.10.0.2 (Graphpad Software, IBM SPSS Inc., Boston, MA, USA). Comparisons of the means were made between groups using un-paired Student’s *t*-tests in instances in which comparing two variables (i.e., no TBI vs. rmTBI, no PTSD vs. PTSD). Analysis of variance (ANOVA) is a statistical model used to compare the variation both between and among groups and was used in this study when comparing three or more variables (i.e., measured depression levels). nQuery was used for power analysis on VLDLR-AS1 to determine significance. *p*-values < 0.05 were considered significant. Pearson correlations were used to examine relationships between lncRNA levels and neuropsychological measures, such as NSI22_tot symptoms, to identify any correlations that could be used as diagnostic indicators for the symptoms. In the ddPCR and qPCR assays, each sample was run in triplicate to ensure reproducibility of results. The optimal cutoff point, measured as AQ in ng for VLDLR-AS1 levels, was calculated using Cutoff Finder [103].

## Figures and Tables

**Figure 1 ijms-25-01473-f001:**
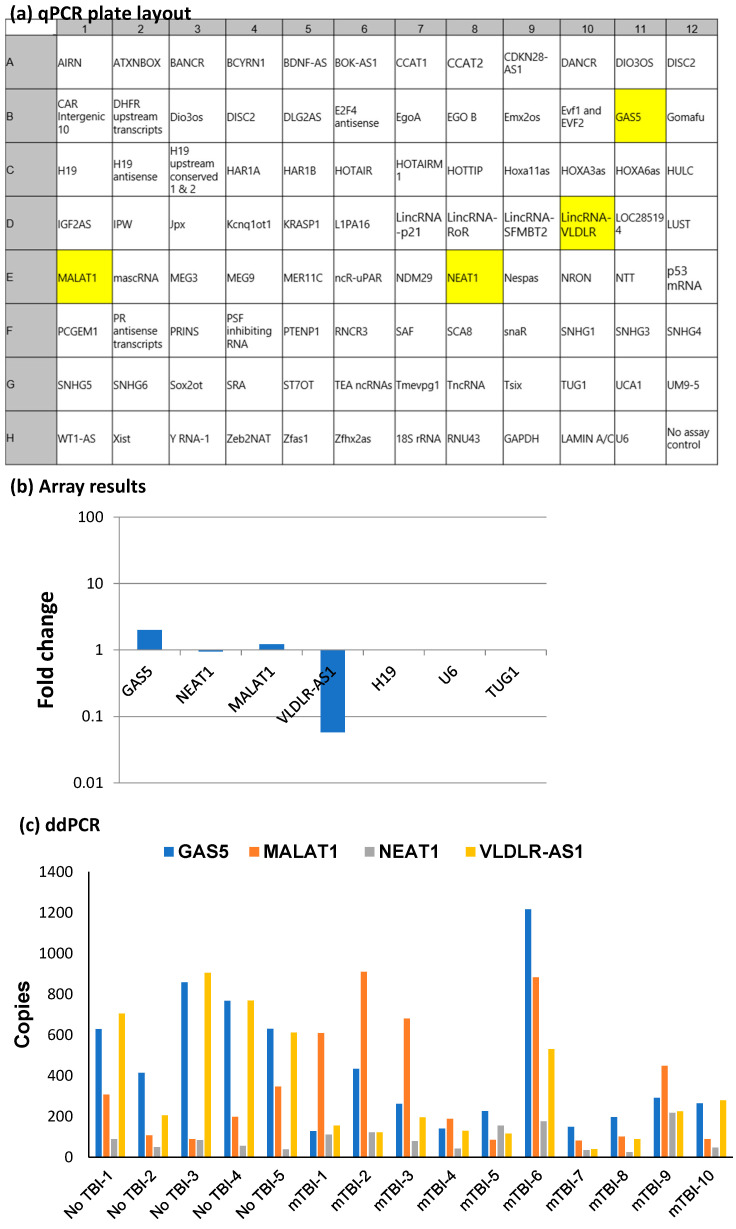
(**a**) The lncRNA profiler array was used to screen the serum lncRNA expression from pooled no TBI (n = 5) and rmTBI (n = 10). The array plate arrangement schematic shows the measurable lncRNAs marked as yellow cells. (**b**) Results from the lncRNA profiler array. Log graph shows fold change calculated as 2^ − (Ct(_av Nor_) − Ct (_av rmTBI_) × Ct(_geo mean_)), where Ct(_avNor_) is the ΔΔCt value of each lncRNA in the no TBI sample; Ct (_av rmTBI_) is the ΔΔCt value of each lncRNA in the rmTBI sample and Ct(_geo mean_) is the ΔΔCt value of the geometric average of multiple internal controls. (**c**) RNA was isolated from individual serum samples from five participants with no lifetime mTBI and from 10 participants with rmTBI and analyzed individually for absolute gene copies using ddPCR. Each sample was run in ddPCR and copies were calculated per mL of serum. The graph shows the absolute gene copies for lncRNAs GAS5, NEAT1, MALAT1 and VLDLR-AS1 in each serum sample.

**Figure 2 ijms-25-01473-f002:**
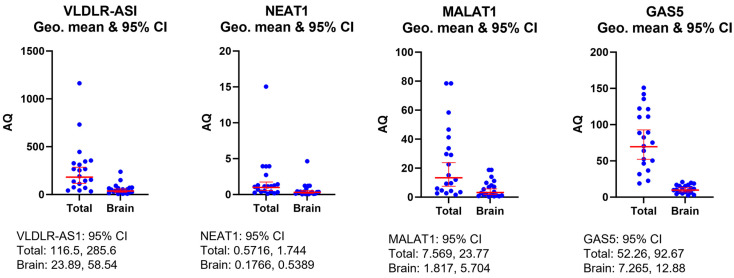
Exosomes derived from CNS were isolated using the L1CAM antibody for pull-down during exosome purification. RNA was isolated from CNS-derived exosomes (brain) and from serum (total), and qPCR was performed using primers specific for lncRNAs MALAT1, GAS5, NEAT1 and VLDLR-ASI. Absolute quantities were determined by normalizing to U6. Statistical analysis was performed in GraphPad Prism SPSS Analysis Software V.10.0.2 to calculate the geo mean 95% CI in both total and brain samples for VLDLR-AS1, NEAT1, MALAT1 and GAS5 levels.

**Figure 3 ijms-25-01473-f003:**
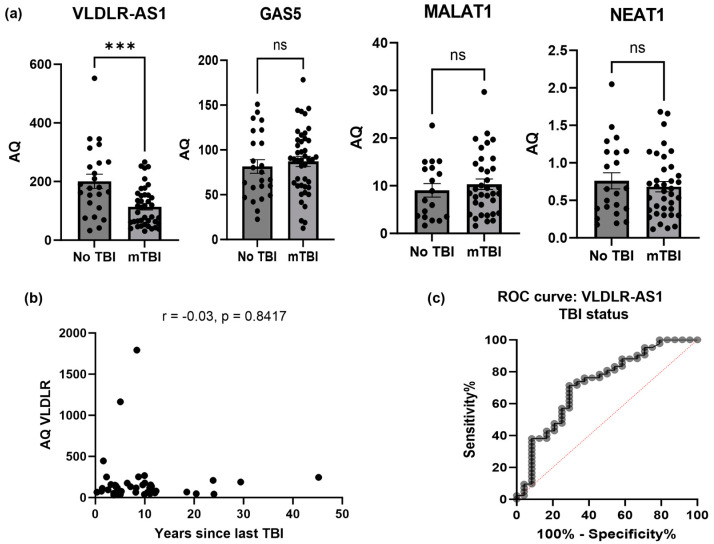
RNA was isolated from serum from individual participants with no lifetime mTBI and from participants with rmTBI. Real-time QPCR was performed using primers specifically for VLDLR-AS1, MALAT1, GAS5, NEAT1 and U6. A standard curve for each lncRNA was included with the qPCR assay, along with samples run in triplicate. (**a**) Absolute quantities (AQs, ng) for each lncRNA were calculated by normalizing the values to U6 absolute levels, followed by calculations for AQ per mL of serum. Statistical analysis was performed using unpaired *t*-tests between no TBI and rmTBI, *** *p* < 0.001, ns: not significant. (**b**) The correlation between levels of VLDLR-AS1 and years since last TBI was evaluated using Pearson’s coefficient and was determined to be r = −0.03, *p* = 0.8417; not significant. (**c**) Receiver operating curve (ROC) was performed on VLDLR-AS1 levels in serum from no TBI and mTBI to determine the optimal cutoff values. Area under curve (AUC) for VLDLR-AS1 is 0.7432 (95% CI: 0.6124, 0.8741); *p* = 0.0012.

**Figure 4 ijms-25-01473-f004:**
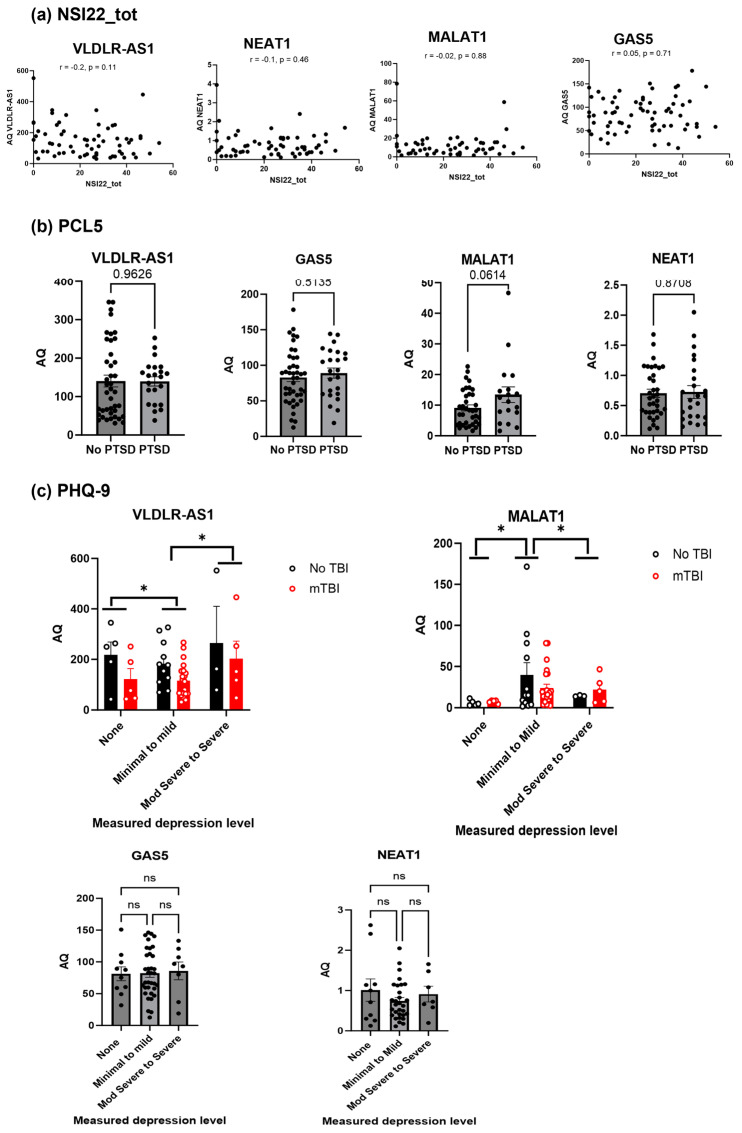
(**a**) The graph shows AQ of each lncRNA versus NSI22_tot score. Correlations between levels of these lncRNAs andNSI22_tot were evaluated using Pearson’s coefficient and were determined not significant for each lncRNA. (**b**) The graph shows AQ of each lncRNA versus the no PTSD and PTSD groups. Statistical analysis was performed using unpaired *t*-tests and was determined not significant for each lncRNA when comparing the no PTSD and PTSD groups. (**c**) The graph shows AQ of each lncRNA versus depression severity level grouped into none, minimal to mild and moderately severe to severe. Statistical analysis was performed using ANOVA to compare none vs. minimal to mild depression level groups and minimal to mild vs. moderately severe to severe depression level groups, * *p* < 0.05, ns = not significant.

**Table 1 ijms-25-01473-t001:** Participant demographics and characteristics.

Characteristic	Study Group	
	No TBI (No. = 24)	mTBI (No. = 43)
Age, mean (SD) (year)	41.0 (12.7)	41.0 (12.7)
Male, No. (%)	19 (79.2)	34 (79.1)
Racial background, No. (%)		
White	16 (66.6)	33 (76.7)
Black	8 (33.3)	10 (23.3)
Education, No. (%)		
High school graduate or GED	3 (12.5)	2 (4.7)
Some college or technical training	9 (37.5)	22 (51.2)
College graduate or higher	12 (50.0)	19 (44.2)
Number of TBI, mean (SD)	0	2.8 (2.1)
Number of blast TBI, mean (SD)	0	0.8 (1.0)
Number of general TBI, mean (SD)	0	2.0 (1.6)
Years since first TBI, mean (SD)		20.2 (12.2)
Years since last TBI, mean (SD)		9.6 (8.5)
PHQ-9 total, mean (SD)	4.0 (4.6)	8.0 (5.6)
PCL-5 total, mean (SD)	14.2 (15.4)	26.5 (16.5)
NSI, mean (SD)		
NSI total	13.3 (13.1)	28.1 (13.3)
Somatic	3.0 (4.2)	7.4 (4.1)
Affective	5.8 (5.0)	9.9 (5.1)
Cognitive	2.8 (2.8)	6.1 (3.7)
Vestibular	1.0 (1.6)	2.9 (2.2)
Number obese (BMI over 30), No. (%)	11 (46.0)	19 (44.0)

## Data Availability

Data are contained within the article.

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
