# Peer review of "Long Noncoding RNA VLDLR-AS1 Levels in Serum Correlate with Combat-Related Chronic Mild Traumatic Brain Injury and Depression Symptoms in US Veterans"

_ijms, 2024, doi:10.3390/ijms25031473_

Round 1

Reviewer 1 Report

Comments and Suggestions for Authors

In the current study, the authors identified several lncRNAs in human serum, which have the potential to help the diagnosis of traumatic brain injury and related symptoms. The study is novel, and the experimental design is rational. I have a few suggestions here, which I hope can help improve the quality of the manuscript.

1. The quality of Figure 1a needs to be improved. I would suggest the authors add some information about how the “fold change” is calculated.

2. In Figure 1b, the explanation of “percent levels” is very confusing to me. For my understanding, the percent level is calculated as the ratio between absolute quantities of lncRNA in exosome and absolute quantities of the same lncRNA in serum. For MALAT1, only around 25% of MALAT1 in the serum if originated from CNS. If the percentage of all four selected lncRNAs are very low, how can the authors justify that the alterations of lncRNA levels are mainly contributed by the brain, not other organs? Regarding the statistic calculation in Figure 1b, what is the biological relevance?

3. Two different methods are used for quantification of lncRNAs, digital PCR and qPCR. In Figure 2, I believe the copy number of VLDLR-AS1 was also lower in the mTBI group than the no TBI group. In the later section, the author switched to qPCR for quantification of lncRNA. What is the reason for not using copy number as the lncRNA level parameter? If the AQ decreased, the copy number should also decrease for the same lncRNA.

4. Figure 4c upper panel, the statistic test is not clear to me. Please clarify which specific groups are being compared.

5. In Figure 4, the authors used three different types of figures to represent the correlation between lncRNA expression and different symptoms.  I would suggest the authors make this part consistent. The figure panel type in 4c showing VLDLR-AS1 and MALAT1 results are most clear to me.

6. Reference is need at line 46-47, following the sentence “military personnel serving in war conflits….”

7. In the paragraph introducing lncRNAs (line 64-82). I would suggest adding some examples of lncRNAs with neurological functions.

8. The use of abbreviations in the content needs revision. Full name should be given the first time it showed up in the content and abbreviations should be used later.

Comments on the Quality of English Language

N/A

Author Response

We sincerely thank the reviewers for the helpful comments to strengthen our manuscript. We have incorporated all the suggestions in the revised manuscript from the reviewers and re-written the sections to clarify the concerns. While the findings remain the same, the manuscript has additional analysis and has been extensively edited.

Reviewer 1: Here is the point-by-point response to the comments:

  1. The quality of Figure 1a needs to be improved. I would suggest the authors add some information about how the “fold change” is calculated.

We have re-written the accompanying result text for Figure 1. Figure 1a now shows a schematic of the lncRNA array plate layout. Figure 1 b is larger and shows the array results as fold change. We have explained the array and how the fold change is calculated in the result section. These are lines 153-194 of the revised manuscript.

  1. In Figure 1b, the explanation of “percent levels” is very confusing to me. For my understanding, the percent level is calculated as the ratio between absolute quantities of lncRNA in exosome and absolute quantities of the same lncRNA in serum. For MALAT1,only around 25% of MALAT1 in the serum if originated from CNS. If the percentage of all four selected lncRNAs are very low, how can the authors justify that the alterations of lncRNA levels are mainly contributed by the brain, not other organs? Regarding the statistic calculation in Figure 1b, what is the biological relevance?

We have re-written the manuscript and this is now result section 2.2 in which we also describe the method and calculation. We have provided the statistical analysis and included the Geo mean 95%CI analysis for biological relevance. This is now Figure 2 (lines 197-233) of the revised manuscript. Further, we also discuss the sources of lncRNAs and relevance to CNS-derived exosomes to measure rmTBI correlation in the discussion section (lines 414-431).

  1. Two different methods are used for quantification of lncRNAs, digital PCR and qPCR. In Figure 2, I believe the copy number ofVLDLR-AS1 was also lower in the mTBI group than the no Ingroup. In the later section, the author switched to qPCR for quantification of lncRNA. What is the reason for not using copy number as the lncRNA level parameter? If the AQ decreased, the copy number should also decrease for the same lncRNA.

ddPCR is now Figure 1c. We have clarified the use of ddPCR to validate the array results and then use of qPCR to evaluate all serum samples for the levels of the four lncRNAs and U6. The results of ddPCR concur with the qPCR results. The ddPCR technology requires state-of-the-art equipment and highly trained personnel and would not be easily incorporated into a clinical setting for screening patients. On the other hand, it is possible to introduce pre-made plates with VLDLR-AS1 primers for use in qPCR thereby requiring minimal technical expertise to facilitate easier translation to clinic. We have included this rationale in the discussion section (lines361-378).

  1. Figure 4c upper panel, the statistic test is not clear to me. Please clarify which specific groups are being compared.

We have re-written this section explaining the grouping and the statistical analysis in Figure 4a,b and c. These are lines 286-332 of the revised manuscript.

  1. In Figure 4, the authors used three different types of figures to represent the correlation between lncRNA expression and different symptoms. I would suggest the authors make this part consistent. The figure panel type in 4c showing VLDLR-AS1 and MALAT1results are most clear to me.

We have re-written the result section 2.4 explaining the grouping and the statistical analysis in Figure 4a,b and c. These are lines 286-332 of the revised manuscript.

  1. Reference is need at line 46-47, following the sentence “military personnel serving in war conflicts….”

We have re-written the introduction and provided the references. These are lines 46-76.

  1. In the paragraph introducing lncRNAs (line 64-82). I would suggest adding some examples of lncRNAs with neurological functions.

We have re-written the introduction and provided the references. These are lines 77-100.

  1. The use of abbreviations in the content needs revision. Full name should be given the first time it showed up in the content and abbreviations should be used later.

We have edited the manuscript and the abbreviations are corrected.

Reviewer 2 Report

Comments and Suggestions for Authors

The article titled "Long noncoding RNA VLDLR-AS1 levels in serum correlate with combat-related chronic mild traumatic brain injury and depression symptoms in US veterans" presents a study investigating the relationship between the levels of the long noncoding RNA VLDLR-AS1 in serum and the symptoms of chronic mild traumatic brain injury (mTBI) and depression in U.S. veterans. The research team used transcriptomic approaches to identify specific lncRNAs in serum, correlating their presence with mTBI.

Based on the content and structure of the article, here are my suggestions for improvement:

1.       The study's scope, particularly the demographic representativeness of the sample, needs clearer articulation, emphasizing the limitations regarding the gender, age, and ethnic diversity of the participants.

Lines 71 – 82: ”There exists a need for a genetic biomarker which can be measured in a non-invasive manner using body fluids to identify patients with mTBI who may benefit from personalized treatment plans in the long term. Blood and urine collections are possible at the clinic and can be done on an annual basis. This study sought to determine a lncRNA expression signature in serum samples that correlates to repetitive mild TBI compared to no lifetime TBI events. Further, a secondary analysis of clinical data of LIMBIC CENC collected as part of a prospective, longitudinal study of mTBI was evaluated for symptom burden such as cognition, memory, depression or PTSD concomitant with mild TBI in the participants. Our results demonstrate that lncRNA VLDLR-AS1 levels are negatively corelated with rmTBI and can serve as a biomarker for diagnosis and prognosis of rmTBI. Identification of VLDLR-AS1 as a genetic biomarker will promote development of novel therapeutics to decrease the chronic burden associated with rmTBI.”

First, there are needed refferences. Second, explain how ”a genetic biomarker which can be measured in a non-invasive manner using body fluids to identify patients with mTBI who may benefit from personalized treatment plans in the long term.”; how ” Identification of VLDLR-AS1 as a genetic biomarker will promote development of novel therapeutics to decrease the chronic burden associated with rmTBI.”

Lines 64-66: ”Long noncoding RNA (lncRNA) are transcription products of genes and do not code for functional proteins. LncRNAs are greater than 200 nucleotides in length and regulate gene expression in normal and disease states” – Which is the source and how can be explained the presence of lncRNA?

2.       Provide more detailed descriptions of the methodologies used, particularly the transcriptomic approaches and statistical analyses, to enhance reproducibility and scientific rigor.

3.       Further elucidation on the biological role and mechanisms of VLDLR-AS1 in relation to brain injury and depression would deepen the understanding of its biomarker potential.

4.       Incorporating comparative analyses with other known biomarkers of mTBI and depression could position the findings within a broader context and highlight the unique contributions of this study.

5.       Expand the discussion to include more comprehensive comparisons with existing literature, addressing how this study's findings corroborate or challenge previous research.

6.       More thoroughly discuss potential confounding variables and how they might affect the study's outcomes, enhancing the robustness of the conclusions.

7.       Improve the clarity and informational value of graphical data representations to make the results more accessible and interpretable to a wider scientific audience.

8.       Suggest a longitudinal study design in future work to track changes in lncRNA levels over time and their correlation with the progression or improvement of symptoms.

9.       Elaborate on the ethical considerations and potential clinical applications of using lncRNA as a biomarker, considering both the benefits and risks.

10.    Provide clear recommendations for future research, including potential experimental designs, to investigate the causal relationships and therapeutic implications of the findings.

Author Response

Reviewer 2: We sincerely thank the reviewer for the helpful comments to strengthen our manuscript. We have incorporated all the suggestions in the revised manuscript from the reviewers and re-written the sections to clarify the concerns. While the findings remain the same, the manuscript has additional analysis and has been extensively edited.

Reviewer 2: Here is the point-by-point response to the comments:

  1. The study's scope, particularly the demographic representativeness of the sample, needs clearer articulation, emphasizing the limitations regarding the gender, age, and ethnic diversity of the participants.

We have re-written the introduction to clarify the scope of the study. Further, we have clearly stated the demographic makeup and characteristics of the participants in result section 2.1 (lines 124-136) as well as in the method section 4.1 (lines 522-549). We have also expanded the limitation section (paragraph following the discussion section- lines 501-519) to address this.

  1. Lines 71 – 82: ”There exists a need … with rmTBI.”

First, there are needed references. Second, explain how ”a genetic biomarker which can be measured in a non-invasive manner using body fluids to identify patients with mTBI who may benefit from personalized treatment plans in the long term.”; how ” Identification of VLDLR-AS1 as a genetic biomarker will promote development of novel therapeutics to decrease the chronic burden associated with rmTBI.”

We have re-written the introduction section to clearly explain the scope of the study, prior research and gaps in the field. We have included references for the section. These are lines 43-121 in the revised manuscript.

  1. Lines 64-66: ”Long noncoding RNA (lncRNA) …disease states” Which is the source and how can be explained the presence of lncRNA?

As stated above, we have re-written the introduction section to clearly explain the scope of the study, prior research and gaps in the field. We have included references for the section. These are lines 43-121 in the revised manuscript.

  1. Provide more detailed descriptions of the methodologies used, particularly the transcriptomic approaches and statistical analyses, to enhance reproducibility and scientific rigor.

We have re-written the results and method section to provide description of the methods and the statistical analyses in the results section 2 and method section 4.

  1. Further elucidation on the biological role and mechanisms of VLDLR-AS1 in relation to brain injury and depression would deepen the understanding of its biomarker potential.

VLDLR-AS1 is expressed in the brain (Harmonizome 3.0) but its function in the pathology or mechanism in brain injury is not known. VLDLR-AS1 is transcribed as an antisense RNA on the opposite strand of the VLDLR gene. Though such naturally occurring antisense RNA may modulate the expression of the sense gene, such as in the case of HMGA2 in pancreatic cancer, currently it is unknown if VLDLR-AS1 affects the function of VLDLR in the brain or in response to injury. Additionally, roles VLDLR-AS1 in ovarian cancer and adipose are discussed. We have included this in the discussion section lines 387-402.

  1. Incorporating comparative analyses with other known biomarkers of mTBI and depression could position the findings within a broader context and highlight the unique contributions of this study.

In the discussion section, we have compared the known protein biomarkers and imaging biomarkers with our study. We include a broader discussion and highlight the unique findings of this study. These are lines 414-444 in the revised manuscript.

  1. Expand the discussion to include more comprehensive comparisons with existing literature, addressing how this study’s findings corroborate or challenge previous research.

We have completely edited our discussion section (lines 345-499) to incorporate existing literature and the context of our study findings.

  1. More thoroughly discuss potential confounding variables and how they might affect the study's outcomes, enhancing the robustness of the conclusions.

We have edited our discussion section (lines 483-491) and expanded the limitation section (lines 501-519) to include confounding variables.

  1. Improve the clarity and informational value of graphical data representations to make the results more accessible and interpretable to a wider scientific audience.

We have edited our result section to explain the methods, analysis and graphs clearly.

  1. Suggest a longitudinal study design in future work to track changes in lncRNA levels over time and their correlation with the progression or improvement of symptoms.

We have included a discussion on the future studies and suggested a study design for use of VLDLR-AS1 as a prognostic biomarker (lines 473-482, 483-491).

  1. Elaborate on the ethical considerations and potential clinical applications of using lncRNA as a biomarker, considering both the benefits and risks.

We have expanded our discussion section for potential clinical applications with pros and cons. Ethically, measuring lncRNA levels is similar to measuring blood glucose or creatinine for diagnosis of disease. It does not involve individual’s whole genome sequencing or personal genetic data.

  1. Provide clear recommendations for future research, including potential experimental designs, to investigate the causal relationships and therapeutic implications of the findings.

We have expanded our discussion section to include this thought and suggestions for future research throughout the discussion section. These are lines 345-499 in the revised manuscript.

Reviewer 3 Report

Comments and Suggestions for Authors

-        The introduction is concise but lacks a comprehensive background on the significance of lncRNAs, especially VLDLR-AS1 and MALAT1, in neurological disorders like TBI and depression. Include more information on existing literature, highlighting the relevance of lncRNAs as potential biomarkers or contributors to these conditions.

-        Clarify the methodological approach used for identifying the four lncRNAs initially identified and their connection to rmTBI.

-        Elaborate on the unbiased transcriptomics approach employed and its specificity in capturing lncRNAs associated with CNS origin within exosomes.

-        Discuss the rationale behind selecting VLDLR-AS1 for further analysis among the initially identified -lncRNAs.

-        Discuss any limitations in the methodology or potential confounding factors that might have influenced the observed results.

-        Offer more details on the correlation between VLDLR-AS1 and MALAT1 with depression symptoms. How strong is this correlation? Is it exclusive to veterans with rmTBI?

-        Expand the discussion to include the clinical implications of using VLDLR-AS1 as a genetic biomarker for rmTBI and depression. How might this discovery impact diagnosis, treatment, or prognosis for affected veterans?

-        Address the limitations of the study. Are there any constraints in using VLDLR-AS1 as a sole biomarker? What are the next steps for further validation or clinical translation?

-        Clarify the rationale behind selecting VLDLR-AS1 for investigation among other potential lncRNAs. Was it based on prior research or preliminary studies suggesting its relevance in TBI?

-        Expand the discussion on the clinical relevance and potential implications of identifying VLDLR-AS1 as a biomarker for rmTBI. How might this biomarker aid in clinical decision-making, prognosis, or therapeutic strategies?

-        Address the limitations of the study thoroughly. Are there any shortcomings in using VLDLR-AS1 as a sole biomarker? What are the implications for generalizing these findings to broader populations of veterans?

-        The use of the lncRNA profiler qPCR Array to identify consistently detected lncRNAs (MALAT1, GAS5, NEAT1, and VLDLR-AS1) in serum samples is well-described. However, more clarity on the rationale behind choosing these specific lncRNAs for evaluation would strengthen the methodology.

-        The findings regarding decreased levels of VLDLR-AS1 in rmTBI compared to no lifetime TBI samples are intriguing. Emphasize the clinical significance of these findings in terms of diagnostic potential and prognostic relevance for identifying individuals with a history of rmTBI.

-        The lack of correlation between VLDLR-AS1 levels and the number of years since the TBI event is an interesting observation. However, discuss potential factors that might contribute to this lack of correlation, such as the persistence of the biomarker over time or sample size limitations.

-        The ROC analysis demonstrating an AUC of 0.74 for VLDLR-AS1 indicates promising diagnostic potential. Elaborate on the interpretation of this value in clinical terms and discuss its utility in practical diagnostic settings.

-        While there's a clear association of lncRNAs VLDLR-AS1 and MALAT1 with depression symptoms, the lack of significant correlations with PTSD or the Neurobehavioral Symptom Inventory-22 (NSI-22) should be discussed. It might be beneficial to consider potential reasons for these discrepancies or non-correlations.

-        Elaborate on the clinical implications of the observed correlation between lncRNAs and depression. How could this association aid in diagnosis, prognosis, or therapeutic interventions for depression among individuals with a history of mTBI?

-        The discussion section provides a comprehensive summary of the study findings and their potential implications.

-        Discuss the significance of the identified lncRNAs (VLDLR-AS1 and MALAT1) in the context of known TBI-associated pathways or processes such as inflammation, neuronal repair, or synaptic plasticity. This can add depth to the discussion of their potential roles in TBI pathology.

-        Consider explaining the rationale behind selecting these specific exclusion criteria and discuss how these criteria ensure homogeneity within the studied population.

-        The statistical methods used are appropriate for the analysis. However, consider providing more details about the statistical software used for specific analyses and mention the rationale behind choosing certain statistical tests over others.

-        Clarify the rationale behind the number of repetitions (3-5 times) for each sample in triplicate. Additionally, discuss the statistical power and sample size calculations performed to ensure adequate power to detect significant differences.

Comments on the Quality of English Language

Minor editing of English language required

Author Response

Reviewer 3: We sincerely thank the reviewer for the helpful comments to strengthen our manuscript. We have incorporated all the suggestions in the revised manuscript from the reviewer and re-written the sections to clarify the concerns. While the findings remain the same, the manuscript has additional analysis and has been extensively edited.

Here is the point-by-point response to the comments:

  • The introduction is concise but lacks a comprehensive background on the significance of lncRNAs, especially VLDLR-AS1 andMALAT1, in neurological disorders like TBI and depression. Include more information on existing literature, highlighting the relevance of lncRNAs as potential biomarkers or contributors to these conditions.

We have re-written the introduction to clarify the scope of the study and the significance of lncRNAs as biomarkers. These are lines 43-121 in the revised manuscript. We have also included the discussion of known roles of VLDLR-AS1 and MALAT1 in the discussion section too.

  • Clarify the methodological approach used for identifying the four lncRNAs initially identified and their connection to rmTBI.

We have re-written the results and method section to provide description of the methods. This is an initial study considered to be the “discovery” study for identifying lncRNAs that differentiate between rmTBI and no TBI. We screened functionally significant lncRNAs in humans and then correlated the quantities to the incidences of mTBI in patients. In addition, in the discussion section we have discussed the role of lncRNAs in TBI (lines 444-464).

  • Elaborate on the unbiased transcriptomics approach employed and its specificity in capturing lncRNAs associated with CNS origin within exosomes.

As stated above, we have re-written the results and method section to provide description of the methods. In the introduction and discussion section, we have included the advantages of lncRNAs as blood-biomarkers of disease. These are lines 77-100 and 494-499 in the revised manuscript.

  • Discuss the rationale behind selecting VLDLR-AS1 for further analysis among the initially identified -lncRNAs.

We have re-written result section 2.3 to clearly explain the rationale and the statistical analysis undertaken for selecting VLDLR-AS1 as the biomarker.

  • Discuss any limitations in the methodology or potential confounding factors that might have influenced the observed results.

We have expanded the limitations section (after discussion section, lines 501-519) to discuss the confounding factors and future studies.

  • Offer more details on the correlation between VLDLR-AS1 and MALAT1 with depression symptoms. How strong is this correlation? Is it exclusive to veterans with rmTBI?

Though our sample cohort consisted of veteran population, we believe the results of this study may be applied to the civilian population too. To do so, in the future we will set up a validation study with larger sample size across the population. We have discussed this aspect in the re-written discussion section. These are lines 465-494 in the revised manuscript.

  • Expand the discussion to include the clinical implications of using VLDLR-AS1 as a genetic biomarker for rmTBI and depression. How might this discovery impact diagnosis, treatment, or prognosis for affected veterans?

We have re-written our discussion section to include the clinical implications of our finding and its application in diagnosis and prognosis of rmTBI. This is expressed throughout the discussion section lines 344-499.

  • Address the limitations of the study. Are there any constraints in using VLDLR-AS1 as a sole biomarker? What are the next steps for further validation or clinical translation?

We have expanded the limitation section as well as re-written the discussion to include the validation studies. These are lines 473-482 in the revised manuscript.

  • Clarify the rationale behind selecting VLDLR-AS1 for investigation among other potential lncRNAs. Was it based on prior research or preliminary studies suggesting its relevance in TBI?

As stated above, this is an initial study considered to be the “discovery” study for identifying lncRNAs that differentiate between rmTBI and no TBI. To be unbiased, we screened lncRNAs and then correlated the quantities to the state of TBI. In addition, in the discussion section we have discussed the connection of lncRNAs to TBI. These are lines 431-444 in the revised manuscript.

  • Expand the discussion on the clinical relevance and potential implications of identifying VLDLR-AS1 as a biomarker for rmTBI. How might this biomarker aid in clinical decision-making, prognosis, or therapeutic strategies?

We have expanded and re-written the discussion section to include these aspects. These are lines 464-499 in the revised manuscript.

  • Address the limitations of the study thoroughly. Are there any shortcomings in using VLDLR-AS1 as a sole biomarker? What are the implications for generalizing these findings to broader populations of veterans?

As stated above, we have expanded the limitation section as well as re-written the discussion to include these aspects. These are lines 465-482, 501-519 in the revised manuscript.

  • The use of the lncRNA profiler qPCR Array to identify consistently detected lncRNAs (MALAT1, GAS5, NEAT1, and VLDLR-AS1) in serum samples is well-described. However, more clarity on the rationale behind choosing these specific lncRNAs for evaluation would strengthen the methodology.

We have re-written the results and method section to provide description of the methods and the significance from the statistical analyses. These are lines 123-172 in the revised manuscript.

  • The findings regarding decreased levels of VLDLR-AS1 in rmTBI compared to no lifetime TBI samples are intriguing. Emphasize the clinical significance of these findings in terms of diagnostic potential and prognostic relevance for identifying individuals with a history of rmTBI.

We have included a thorough discussion of these aspects in the revised manuscripts. This is throughout the discussion section lines 344-499.

  • The lack of correlation between VLDLR-AS1 levels and the number of years since the TBI event is an interesting observation. However, discuss potential factors that might contribute to this lack of correlation, such as the persistence of the biomarker over time or sample size limitations.

We believe that these observations can be pursued with a validation study with a larger sample size as well as analyzing the levels of VLDLR-AS1 immediately after a mTBI event and the annually in a longitudinal study. We have included this in the re-written discussion section. These are lines 483-491 in the revised manuscript.

  • The ROC analysis demonstrating an AUC of 0.74 for VLDLR-AS1indicates promising diagnostic potential. Elaborate on the interpretation of this value in clinical terms and discuss its utility in practical diagnostic settings.

In the re-written discussion section, we have included the current state of using lncRNAs as biomarkers in disease. The importance of this finding is that had a strong correlation coefficient and high significance in statistical analysis in populations where mTBI was clinically confirmed with imaging and neuropsychological tests. These are lines 464-499 in the revised manuscript.

  • While there's a clear association of lncRNAs VLDLR-AS1 and MALAT1 with depression symptoms, the lack of significant correlations with PTSD or the Neurobehavioral Symptom Inventory-22 (NSI-22) should be discussed. It might be beneficial to consider potential reasons for these discrepancies or non-correlations.

We propose, in the discussion section, to expand this finding to a validation study with a larger number of samples in the future. This may provide insight for the observed non-correlations. These are lines 431-444 in the revised manuscript.

  • Elaborate on the clinical implications of the observed correlation between lncRNAs and depression. How could this association aid in diagnosis, prognosis, or therapeutic interventions for depression among individuals with a history of mTBI?

We have re-written our discussion section to include the clinical implications of our finding and its application in diagnosis and prognosis of depression. These are lines 445-464 in the revised manuscript.

  • The discussion section provides a comprehensive summary of the study findings and their potential implications.

We have re-written and expanded the entire discussion section.

  • Discuss the significance of the identified lncRNAs (VLDLR-AS1 and MALAT1) in the context of known TBI-associated pathways or processes such as inflammation, neuronal repair, or synaptic plasticity. This can add depth to the discussion of their potential roles in TBI pathology.

The lncRNAs are packed in exosomes to prevent its degradation. We have discussed that MALAT1 is implicated in TBI while analyzing brain tissue. However, determination of MALAT1 serum values was not undertaken. VLDLR-AS1 has not yet been implicated in TBI and we have discussed potential roles and methods that may be undertaken in the future to determine the function of VLDLR-AS1 in TBI. These are lines 379-444 in the revised manuscript.

  • Consider explaining the rationale behind selecting these specific exclusion criteria and discuss how these criteria ensure homogeneity within the studied population.

We have expanded the description of clinical sample selection (lines 522-549). We excluded all patients with any form of cancer as lncRNAs are shown to be upregulated in cancer.

  • The statistical methods used are appropriate for the analysis. However, consider providing more details about the statistical software used for specific analyses and mention the rationale behind choosing certain statistical tests over others.

We have expanded the result and the method sections to include the rationale for the statistical analysis and the choice of tests appropriate for evaluation of the correlation in the different groups. These are lines 626-638 in the revised manuscript.

  • Clarify the rationale behind the number of repetitions (3-5 times) foreach sample in triplicate. Additionally, discuss the statistical power and sample size calculations performed to ensure adequate power to detect significant differences.

We have clarified that the repetitions are used in the qPCR and ddPCR analysis. We discuss the statistical tests and sample size calculations in the revised manuscript. These are lines 626-638 in the revised manuscript.

Round 2

Reviewer 2 Report

Comments and Suggestions for Authors

The authors have consistently improved the manuscript and responded to all of my previous comments and suggestions. As the responses satisfy my inquiries, I conclude for acceptance in the present form.

Reviewer 3 Report

Comments and Suggestions for Authors

The revised manuscript, with all raised questions answered, is recommended for acceptance.

Comments on the Quality of English Language

Minor editing of English language required